

# Combined assimilation of streamflow and snow water equivalent for mid-term ensemble streamflow forecasts in snow-dominated regions

Jean M. Bergeron[1], Mélanie Trudel[1], Robert Leconte[1]

[1]Department of civil engineering, Université de Sherbrooke, Sherbrooke, J1K 2R1, Canada

*Correspondence to*: Jean M. Bergeron (Jean.Bergeron2@USherbrooke.ca)

**Abstract.** The potential of data assimilation for hydrologic predictions has been demonstrated in many research studies. Watersheds over which multiple observation types are available can potentially further benefit from data assimilation by having multiple updated states from which hydrologic predictions can be generated. However, the magnitude and time span
of the impact of the assimilation of an observation varies according not only to its type, but also to the variables included in the state vector. This study examines the impact of multivariate synthetic data assimilation using the Ensemble Kalman Filter (EnKF) into the spatially distributed hydrologic model CEQUEAU for the mountainous Nechako River located in British-Columbia, Canada. Synthetic data includes daily snow cover area (SCA), daily measurements of snow water equivalent (SWE) at three different locations and daily streamflow data at the watershed outlet. Results show a large variability of the
continuous rank probability skill score over a wide range of prediction horizons (days to weeks) depending on the state vector configuration and the type of observations assimilated. Overall, the variables most closely linearly linked to the observations are the ones worth considering adding to the state vector. The performance of the assimilation of basin-wide SCA, which does not have a decent proxy among potential state variables, does not surpass the open loop for any of the simulated variables. However, the assimilation of streamflow offers major improvements steadily throughout the year, but
mainly over the short-term (up to 5 days) forecast horizons, while the impact of the assimilation of SWE gains more importance during the snowmelt period over the mid-term (up to 50 days) forecast horizon compared with open loop. The combined assimilation of streamflow and SWE performs better than their individual counterparts, offering improvements over all forecast horizons considered and throughout the whole year, including the critical period of snowmelt. This highlights the potential benefit of using multivariate data assimilation for streamflow predictions in snow-dominated regions.

**Keywords**. Ensemble Kalman Filter, hydrological modeling, ensemble forecast, snow accumulation and melt, multivariate data assimilation





# 1 Introduction

Water resource management for reservoirs located in snow dominated regions relies on an accurate portrayal of the snow water equivalent (SWE) spatial and temporal distribution in order to make accurate streamflow predictions. Some water resources managers make use of Ensemble Streamflow Prediction (ESP) to plan reservoir operations over various lengths of

time. ESPs have the benefit of integrating weather forecast uncertainty, either by making use of weather ensemble predictions (de Roo et al, 2003) or by using historical weather data (Day, 1985) as input in a hydrologic model. However, ESPs depend heavily on the model's initial conditions (Franz, 2008). Presently, many water resources managers still use a manual approach to adjust the initial state of the watershed based on available observations and the user's experience (Liu et al, 2012).

Data assimilation (DA) methods, such as the Ensemble Kalman Filter (EnKF; Evensen, 2003) can improve the estimation of the initial state of the watershed while also providing an uncertainty on this initial state (Liu and Gupta, 2007). Several authors have already shown the added value of DA in snow-dominated watersheds to improve the estimation of the state of the watershed (De Lannoy et al, 2012; Dechant and Moradkhani, 2011; Nagler et al, 2008; Slater and Clark, 2006; Andreadis

and Lettenmaier, 2006). Some studies have also integrated DA in ensemble forecast systems for relatively short-term (up to 5-10 days) hydrologic forecasts (Abaza et al, 2015; Abaza et al, 2014; He et al., 2012), but studies focusing on longer forecast periods are scarce even though the need exists for water resource managers.

Multivariate DA applications in hydrology are becoming more frequent, but generally focus on streamflow and soil moisture

(Samuel et al, 2014; Trudel et al, 2014; Lee et al, 2011), omitting snow water equivalent. In snow-dominated watersheds, the key initial states include not only information about the hydric state, such as soil moisture and streamflow, but also the snow cover state, such as snow water equivalent (SWE) and snow cover area (SCA). To the authors' knowledge, no published studies pertain to the combined assimilation of information about a watershed's hydric and snow state. Since the lasting impact of hydric DA and snow DA can be quite different given the different physical processes driving them, the

simultaneous DA of both types of data could yield improvements over a potentially longer length of time.

However, data assimilation performance depends on various factors, such as the choice of variables to be updated by an observation (hereby referred to as the state vector configuration). Abaza et al. (2015) have demonstrated this importance when assimilating streamflow in a hydrologic model. Going from univariate to multivariate DA increases the number of

degrees of freedom, which increases the complexity of the matter. Is the state vector configuration still important in multivariate DA?



The study's main objectives are to 1) investigate the potential impact that multivariate data assimilation of hydric (streamflow) and snow-related (SWE and SCA) data can have on short-term (1-5 days) and mid-term (up to 50 days) streamflow forecast, and 2) to explore how this impact varies as a function of the state vector configuration.

## 2 Materials and methods

### 2.1 Study area description and data

Simulations were conducted in a synthetic setting based on the Nechako watershed located in British-Columbia, Canada (Fig. 1). The watershed includes a reservoir, which drains an area of approximately 14000 km$^2$. The reservoir is managed by Rio Tinto mainly for hydroelectricity production purposes. The watershed includes part of the Coast Mountains in the west region, such that the difference in elevation reaches about 1700 m. At these latitude and altitude, most of the precipitation falls as snow.

There are various types of data gathered regularly over the watershed. First are seven weather stations managed by Rio Tinto, three of which measure daily precipitation and air temperature only (yellow circles). Three others also include snow pillows (red squares), which measure snow water equivalent. The northernmost snow pillow is located at Mount Wells, the southernmost at Mount Pondosy and the westernmost at Tahtsa Lake. Maximum seasonal SWE observations average 615, 853 and 1393 mm for the Mount Wells, Mount Pondosy and Tahtsa Lake snow pillows respectively. The distribution of snow on the ground follows a strong East-West gradient such that measurements at Tahtsa Lake typically yield much more snow that Mount Well and Mount Pondosy. The northernmost weather station (blue triangle) is located next to the spillway at Skins Lake and also takes hydrometric measurements. Historical daily water levels can then be converted into natural inflows by also taking into account spilled and turbined flow. Finally, daily snow cover area (SCA) data derived from the spaceborne sensor MODIS/Terra are also considered (Hall et al, 2002). Because of its spatial coverage and relatively high temporal resolution, remotely sensed snow data from MODIS have proven to be valuable in a number of hydrologic studies (Bergeron et al, 2014; Roy et al, 2010; Tang and Lettenmaier, 2010; Andreadis and Lettenmaier, 2006; Clark et al, 2006), including one applied to the Nechako watershed (Marcil et al, 2016).

The meteorological observations gathered over a period of 10 year (from 15[th] august 1990 to 14[th] august 2000) will be used to as a basis upon which a synthetic experiment (see below) will test the added value of three types of synthetic observations (streamflow, SWE and SCA) for data assimilation purposes. The only dataset actually used is the meteorological station data, while observation data are created synthetically to mimic streamflow, SWE and SCA data that could be measured or estimated using hydrometric, snow pillow and MODIS data, respectively.



## 2.2 Model description

The hydrologic model used is the spatially distributed, conceptual model CEQUEAU (Charbonneau et al, 1977). It is currently being used by Rio Tinto Alcan to model streamflow at the outlet of the Nechako watershed, considered to be the spillway where the hydrometric station is also located. A summary of the main processes concerning the production and

transfer functions is presented here to facilitate the understanding of the state variables used in this study.

CEQUEAU divides the watershed into regular square pixels called "whole squares" over which the production function is computed Fig. 2). The current version of CEQUEAU uses the snow model presented by the U.S. army corps of engineers (1956) to simulate most snow-related processes. The SWE is actually computed separately for forested and open areas,

which have their own set of parameters, but is aggregated here as a weighted sum according to the proportion of forested and open areas within each whole squares. The only variable computed separately is SCA, which is computed using a depletion curve (Anderson 1973). The depletion curve used here follows Andreadis and Lettenmaier (2006), which uses a three parameter beta distribution:

$$\mathrm{SCA}_i = \mathrm{B}^{-1}\left(\frac{\mathrm{SWE}_i}{\min(\mathrm{SWE}_{\max,i}, SI)}\middle|\alpha_{SCA}, \beta_{SCA}\right),\qquad(1)$$

where $\mathrm{SCA}_i$ is the resulting snow cover area over a whole square $i$, $\mathrm{SWE}_i$ is the simulated snow water equivalent over the same area, $\mathrm{SWE}_{\max,i}$ is the annual maximum snow water equivalent since the beginning of the accumulation period over the same area, $SI$ represents the value of SWE above which it is assumed there is always 100% snow cover and $\alpha_{SCA}$ and $\beta_{SCA}$ are shape parameters for the beta distribution itself. These three parameters were previously calibrated to match MODIS/Terra daily L3 snow cover data (Hall et al, 2002). It is important to note that SCA is computed as an output only and

is therefore not considered to be a state variable since it has no impact on future simulations if its value is tempered with.

CEQUEAU then uses three conceptual reservoirs to simulate various hydrologic processes from the available water resulting from rain or snow melt. There is an optional lake reservoir, an upper reservoir (called "soil moisture reservoir" in this study) and a lower reservoir (called "groundwater reservoir" in this study).

All in all, the state variables simulated over each whole square includes snow water equivalent (SWE), a snow ripening index (SRI), a snow temperature index (STI), the soil moisture level (SML), the groundwater level (GWL) and the lake water level (LWL) should there be one. There are 644 whole squares in the case of the Nechako watershed.

Each whole square is itself divided into "partial squares" according to the subpixel drainage divide. There are a total of 1082 partial squares in the case of the Nechako watershed. Available water from whole squares is divided into these partial





squares to form volumes (VOL) that are transferred from one partial square to the next at a specified rate to form streamflows as follows :

$$SF_j = \frac{1}{\Delta t} \sum_k^{M_j} ext_k \cdot \text{VOL}_k, \tag{2}$$

where $SF_j$ is the streamflow at partial square $j$, $M_j$ is the number of partial squares directly upstream $ext_k$ is a transfer
coefficient and $\Delta t$ is the time step. Streamflow, like SCA, is not considered to be a state variable since it has no impact on future simulations if its value is tempered with.

**2.3 Ensemble Kalman Filtering**

The Ensemble Kalman Filter (EnKF) is a data assimilation method developed by (Evensen 1994). It is an approach often used in hydrology, mainly due to its ability to consider non-linearities in the model and its relative simplicity to implement.
The EnKF is a sequential method, meaning it relies only on current observations to update state variables as opposed to non-sequential approaches such as smoothers (Evensen and van Leeuwen, 2000) and recursive (McMillan et al, 2013) methods.

However, the method has practical and theoretical limitations. Firstly, the EnKF relies on an ensemble representation of model and observation errors that are valid in the limit where ensemble sizes approach infinity. This is not feasible in
practice, so a finite sample is used instead which aims to be sufficiently large such that sampling errors are negligible while ensuring that computational power and memory limitations are met. The method also makes use of model and observation covariance matrices to compute the gain during the updating process. These covariance matrices assume a linear relationship between variables. Inaccurate analyses may result for cases where this assumption does not hold. Despite these limitations, the EnKF's usefulness in hydrology has nonetheless been shown in numerous studies, which is why it is also used in this
study.

The EnKF propagates an ensemble of model runs based on a Monte Carlo implementation to represent model errors. The model covariance matrix ($P_t^b$) at a time $t$ is computed from the state vector ($x_t^b$) holding the $N$ ensemble members and their simulated variables; and the ensemble mean of the state vector ($\overline{x_t^b}$), therefore implicitly taking the model dynamics into
consideration:

$$P_t^b = \frac{1}{N-1} \left( x_t^b - \overline{x_t^b} \right) \left( x_t^b - \overline{x_t^b} \right)^\top, \tag{3}$$

When an observation is available, it is perturbed to form an ensemble of observations that are used to update each ensemble member. The updating step applies the Kalman gain ($K_t$), which is computed from observation ($R_t$) and model covariance matrices as well as an observation operator ($H_t$), which relates the model states to the observation:

$$K_t = P_t^b H_t^\top (H_t P_t^b H_t^\top + R_t)^{-1}, \tag{4}$$



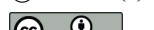

The Kalman gain acts as a weighted average between the observation and state vector to yield a post-filter analysis ($x_t^a$) computed as such:

$$x_t^a = x_t^b + K_t(y_t - H_t x_t^b),\qquad(5)$$

The Ensemble Kalman filter aims to minimize the resulting analysis error under several assumptions such as having
normally distributed, bias-free and time-independent errors, as well as linear relationships between variable errors during the updating step. Though these assumptions are not always met, which means that optimality is not guaranteed, the value of the EnKF for hydrologic applications has nevertheless been shown repeatedly (Abaza et al, 2015; Samuel et al, 2014; Trudel et al, 2014; Brocca et al, 2012; Forman et al, 2012; Xie and Zhang, 2010; Weerts and El Serafy, 2006).

## 3 Experimental design

### 3.1 Synthetic experiment

A synthetic experiment is a useful approach to test the robustness of a data assimilation method or to tune various hyper-parameters. This is because the true state is known since it is initially created from known inputs.

For the current study, interpolated meteorological observations and a specific set of parameters were used to run
CEQUEAU, the output of which was considered to be the true state (1) (see Fig. 3). Synthetic observations (2) and meteorological input (3) were then obtained by applying a perturbation to the true state and true meteorological input respectively. This means that the observations sets described thus far are not directly used, but synthetically generated using known parameters and perturbation. The synthetic observation ensemble (4) and meteorological ensemble (5) were created by further perturbing the synthetic observations and meteorological input. Ideally, these meteorological and observation
ensemble perturbations should reflect the true errors of the synthetic meteorological input and observations for an optimal analysis. An ensemble of hydrologic states (6) was then obtained by running CEQUEAU using the synthetic meteorological ensemble. The EnKF was then applied using both the model and observation ensembles to produce an analysis (7), which was used as an initial state to produce ensemble streamflow predictions (ESP; 8) using the true meteorological input.

Using the true meteorological input implies that over a sufficiently large forecast horizon, every DA scenario considered in this study is likely to converge to the true state, but at different rates. By comparing the relative gains in performance over the ensemble with no data assimilation (open loop), one can then observe the length of time upon which DA impacts ESPs without having erroneous meteorological input affecting the results.



## 3.2 Hyper-parameter tuning

The use of the EnKF requires the tuning of hyper-parameters, such as model and observation errors, and ensemble size. Improper specification of these hyper-parameters could lead to filter divergence (Houtekamer and Mitchell, 1998).

### 3.2.1 Ensemble size

The ensemble size should ideally approach infinity to reduce to impact of sampling when covariance matrices are computed, but this is not feasible given the limits of computing power and memory. In practice, the ensemble size is chosen such that computing time is more reasonable while ensuring that sampling error remains small.

Tests were carried out using ensemble sizes of 8, 16, 32, 64 and 128 members. An ensemble size of 64 members was used
for this study. This number was chosen as a function of the stability between successive runs and computing resources available. It was found to be a reasonable trade-off between having sufficiently consistent results between simulations, such that the sampling error would be dwarfed in comparison with the impact of the actual data assimilation, without exceeding the computing resources available.

### 3.2.2 Meteorological input perturbation

In order to reduce the impact of model error misrepresentation, the perturbation factors used to generate an ensemble spread were the same as the ones used to generate synthetic meteorological inputs. Errors from CEQUEAU-specific parameters were not taken into consideration, such that the parameter set used for the generation of the true state were the same for the ensemble generation.

Both the precipitation and temperature values were perturbed using a gamma distribution, which has the benefit of generating positive values exclusively. Perturbations were implemented such that the weather input $z_{t,i}$ (precipitation or temperature) at time $t$ over the partial square $i$ is the result of the inverse gamma function given the cumulative probability $P_{t,i}$ of a spatially and temporally correlated Gaussian random field with zero mean and unit variance. This can be expressed mathematically as $z_{t,i} \sim \Gamma^{-1}\left(P_{t,i} | \kappa, \theta\right)$, where $\kappa$ and $\theta$ are shape and scale factor respectively. The shape and scale factors can
be expressed in terms of mean $\mu_z$ and variance $\sigma_z^2$, such that $\kappa = \mu_z^2/\sigma_z^2$ and $\theta = \sigma_z^2/\mu_z$. In this study, synthetic precipitations are generated using the value of the true precipitation for $\mu_z$ and a relative variance $\sigma_z^2$ of 50%. Similarly, perturbed temperatures use the true temperatures for $\mu_z$ and an absolute standard deviation of 1°C. Within the synthetic study where the feasibility of the approach is tested, the exact value of these perturbations is arbitrary, so long as it is coherent between scenarios. The values used were such that the performance of the mean streamflow ensemble generated using
CEQUEAU was roughly similar to the performance of the simulated streamflow using real-world meteorological input compared with observations.





### 3.2.3 Synthetic observation error representation

As with model error representation, the perturbation factors used to generate an ensemble of observations were the same as the ones used to generate synthetic observations. Three types of observations are considered; namely streamflow, snow water equivalent (SWE) and snow cover area (SCA).

Synthetic streamflow and SWE observations were created in a similar fashion than synthetic meteorological inputs, but using perturbations that have a lognormal distribution. Observations $y_{t,i}$ can be expressed as $y_{t,i} \sim L^{-1}(Q_{t,i} | \mu_y, \sigma_y^2)$, where $Q_{t,i}$ is the cumulative probability of a temporally correlated Gaussian random field with zero mean and unit variance. The true state was used for $\mu_y$ for both streamflow and SWE, while the relative variance $\sigma_y^2$ was set to 20 % and 10 % respectively. As

with the meteorological input error, the exact value of these variances is arbitrary for feasibility purposes. However, since the conclusions of this study will likely be used to help setup real-world applications, the variances chosen should ideally be relatively similar to the error of their corresponding real observations. Since these real observation errors are not known, rough estimates are used.

Synthetic watershed-wide SCA were created using perturbations that follow a beta distribution since SCA is bounded between 0 and 1. SCA observations are expressed as $y_{t,i} \sim B^{-1}(Q_{t,i} | \alpha, \beta)$, where $\alpha$ and $\beta$ are positively valued shape parameters. The shape parameters may be expressed in terms of $\mu_y$ and $\sigma_y^2$, but it must follow that $\sigma_y^2 < \mu_y(1 - \mu_y)$. The variance was arbitrarily set to $\sigma_y^2 = \mu_y(1 - \mu_y)/50$, such that the resulting shape parameters are $\alpha = 49\mu_y$ and $\beta = 49(1 - \mu_y)$. This translates as a null variance when the snow covers either 0 or 100 % of the watershed and a variance

greatest at 50 % SCA. This prevented introducing a systematic bias when assimilating SCA values at 0 or 100 %, as well as giving the observations a greater uncertainty during the transition periods when SCA, which loosely follows the greater uncertainty attributed to MODIS observations over the same period (Hall and Riggs, 2007).

### 3.2.4 Covariance localization

The main disadvantage in using a finite sample to compute covariance matrices is that the resulting covariance matrices are

not exact. This may result in theoretically zero covariance elements between two theoretically uncorrelated variables to become small, but non-zero, which may deteriorate the performance of the EnKF.

One way to overcome this dilemma is to use covariance localization, where covariances are forced to zero between some variables. One option is the Schur product where a covariance matrix is multiplied element-wise by a distance-dependant

correlation function (Houtekamer and Mitchell, 2001). However, there are other geophysical characteristics, such as landcover and elevation, which could be considered in the covariance localization. This would further increase the number of parameters to set and the degree of subjectivity in setting those parameters when the degrees of dependence are unknown.





Another approach was used in this study, which is based on the improvements observed in the state vector. First, the open loop is executed, as well as a data assimilation scenario with one observation and the corresponding spatialized state variable included in the state vector (ex: 1 snow pillow assimilated and all modelled SWE included in the state vector). Then, the two runs are compared with the true state on a spatial basis. In the case of CEQUEAU, these can be whole or partial squares depending on the variable analysed. The covariance matrix is localized such that the areas that do not show an improvement for the data assimilation scenario over the open loop are set to zero. This process is repeated for each observation.

While this process remains susceptible to the sampling error from the finite ensemble size, it is a simple approach that exploits the availability of the true state in a synthetic experiment and limits the state vector size according to observed improvements.

In this study, only SWE observations have a corresponding state variable, so covariance localization will only be applied to the SWE variable.

### 3.2.5 State vector configuration

Though the state vector often comprises only of the variables corresponding to the observations or those judged to be relevant enough by the user, there are potentially many state variables which could benefit from the assimilation of available data if there exists a linear (or approximately linear) relationship between the modelled variables and the observations.

To determine which variable could benefit from being included in the state vector, one could execute multiple scenarios where each possible combination is compared with the true state. However, this could get very laborious even for a relatively small number of state variables. The current approach suggests reducing this number by first adding state variables one at a time. The variables which show a global improvement can then be added to the state vector. Assuming that not all variables are added to the state vector, this reduces the number of combinations to try.

### 3.3 Metrics

Various metrics are used to quantify results. The Mean Square Skill Score (MSSS), based on the Mean Square Error (MSE) are used to assess the differences between various data assimilation scenarios and the open loop during the state vector configuration and covariance localization processes. The MSE for a variable of interest $x$ is defined as:

$$\text{MSE}(x) = \frac{1}{N}\sum_{t=1}^{N}(\overline{x_t} - x_t^{\text{T}})^2, \tag{6}$$

where $N$ is the number of time steps, $\overline{x_t}$ is the ensemble mean analysis of the state variable of interest at time $t$ and $x_t^{\text{T}}$ is the corresponding true state. It is often more convenient to express this score as a unitless skill score:



$$\text{MSSS}(x) = 1 - \frac{\text{MSE}(x)}{\text{MSE}_{\text{ref}}(x)}, \qquad (7)$$

where $\text{MSE}_{\text{ref}}(x)$ is a mean square error of reference; the open loop in this case. The MSSS is bounded by $[-\infty, 1]$ and indicates an improvement as the skill score increases. Values above zero indicate an improvement over the reference (open loop) and a value of one indicates a perfect score; a perfect correspondence between the mean of the analysis and the true state.

The ensemble forecast performance is assessed using the Continuous Rank Probability Score (CRPS; Hersbach, 2000) and its associated skill score (CRPSS). For this synthetic study, the CRPS is adapted as follows:

$$\text{CRPS}(x, f) = \frac{1}{N}\sum_{t=1}^{N}\int_{-\infty}^{+\infty}\left(F\left(x_t^f\right) - F(x_t^{\text{T}})\right)^2 \mathrm{d}x, \qquad (8)$$

where $F\left(x_t^f\right)$ and $F(x_t^{\text{T}})$ are the cumulative distribution function of the ensemble forecast at a horizon $f$ and the true state, respectively. The CRPS has the same units as the variable of interest and is bounded by $[0, +\infty]$. A lower CRPS is a better score. As with the MSE and MSS, it is often convenient to express the CRPS in its skill score form:

$$\text{CRPSS}(x, f) = 1 - \frac{\text{CRPS}(x, f)}{\text{CRPS}_{\text{ref}}(x, f)}, \qquad (9)$$

where $\text{CRPS}_{\text{ref}}(x, f)$ is the continuous rank probability score of the open loop used as a reference in this case. Like the MSSS, the CRPSS is bounded by $[-\infty, 1]$, with higher values indicating a better score. Values above zero indicate an improvement over the reference and a value of one indicates a perfect score.

## 4 Results and discussion

### 4.1 State vector configuration and covariance localization

Before investigating the effect of data assimilation on streamflow forecasts, a state vector configuration analysis is conducted. This is done in order find out which variables, among the 7 listed previously (VOL, SWE, SRI, STI, SWL, GWL, LWL), should be included in the state vector for each type of data assimilated in order to reduce the number of comparisons to make.

### 4.1.1 Streamflow data assimilation

First presented are the results from the case where only streamflow at the outlet is assimilated. Streamflow at the outlet is computed by the model, but it is output-only. Therefore, in order for the assimilation of streamflow to have any impact on the modelled states, additional variables must be added to the state vector.





Figure 4 shows a boxplot of the MSSS computed for each variable on the whole watershed when they are individually included in the state vector using the open loop scenario as a reference. Values above 0 means there is an improvement for a particular partial (for volumes) or whole (for other state variables) square compared with the open loop. The boxes range between the 25$^{th}$ and 75$^{th}$ percentiles, with a red bar to show the median, and the whiskers range between the maximum and

minimum values. Outliers are not shown for visibility purposes. Results for the case where water volumes (VOL) are included along with the streamflow at the outlet show an improved score for each partial square on the watershed. This is not entirely surprising given the close relationship between streamflow and volume. This suggests a necessity to include VOL in the state vector when assimilating streamflow at the outlet for streamflow predictions.

Results also show a deterioration of snow water equivalent (SWE) for nearly 75% of whole squares on the watershed. Although there is some improvement for some whole squares, this suggests that including SWE in the state vector when assimilating streamflow at the outlet is unlikely to be beneficial for streamflow predictions. Although SWE does have an important impact on streamflow, there is a time lag between the snowmelt occurrence and the increase of streamflow at the outlet. Since the EnKF assumes linear relationships between variables, the non-linearity between SWE and streamflow can

result in a non-optimal analysis. In this case, the results are actually worse than open loop for most whole squares. Clark et al, (2008) discuss the issue of non-linearities between streamflow and other variables. To overcome this issue, one could use either a recursive approach, which allow adjustments of previously simulated variables, or smoother approach to DA, which also uses "future" observations to update current state variables. However, this may not be necessary given the positive impact of streamflow DA on VOL, as well as in a multivariate DA scenario where other variables, such as SWE in this case,

are also assimilated.

As for the snow ripening index (SRI) and snow temperature index (STI), the median sits around 0, meaning there is no improvement for 50% of the whole squares. This suggests little change can be obtained in the analysis by including those variables in the state vector. Similar to the case with SWE, there is likely a time lag issue between streamflow and these

variables. However, there is also a weaker link between these variables such that a change in SRI or STI is not as strongly linked to an eventual change in streamflow as much as it is for a change in SWE.

Finally, results for the three conceptual reservoirs soil moisture level (SML), groundwater level (GWL) and lake water level (LWL) show an improvement for over half of the whole squares, with a greater number of whole squares improved for the

GWL and slightly above 0 median for SML. This suggests that including these three variables in the state vector can potentially yield improvements for streamflow predictions. Though the relationship between the water level in these conceptual reservoirs and streamflow at the outlet is not exactly linear, mainly due to reservoirs having multiple orifices (see Fig. 2) and the time lag before water reaches the outlet, it may be sufficiently near linear that such that streamflow DA yields an overall improvement for most whole squares. For example, the median correlation coefficient of a simple linear





regression between each reservoir for each whole square and the streamflow at the outlet is 0.12, 0.49 and 0.15 for SML, GWL and LWL respectively.

The inclusion of VOL, SML, GWL and LWL in the state vector will therefore be considered during the assimilation of
streamflow at the outlet. The impact of each scenario for streamflow predictions will be compared.

### 4.1.2 SWE data assimilation

The same analysis is performed for SWE data assimilation from synthetic snow pillows. However, unlike streamflow, SWE is a state variable such that any changes made upon it will have repercussions on future simulations. SWE at the location of the snow pillows should therefore be included in the state vector and also potentially whole squares in the vicinity that are
correlated with these locations. A spatial analysis is performed first to determine the spatial extent upon which each snow pillow may affect modelled SWE in other whole squares.

Figure 5 shows the MSSS of SWE on the spatial level for the snow pillows located at Mount Wells, Mount Pondosy and Tahtsa Lake, using the open loop scenario as a reference. For each figure, the whole square which shows the most
improvement is the area where the corresponding snow pillow is located. Whole squares which show improvements are mainly located around snow pillows, but the range differs for each snow pillows. Various areas in remote locations also show improvements for each snow pillow. As mentioned earlier, relationship with geophysical factors, such as distance from snow pillow, elevation and land cover, could be used to explain this variation, but a simpler approach was used such that the covariance localization was limited to whole squares showing improvements only. The covariance elements representing all
the other whole squares were set to zero.

As for the state vector configuration, Fig. 6 shows the MSSS computed for each variable within the extent of whole squares which were positively impacted by SWE DA during the covariance localization process. The open loop scenario was used again as a reference. The results show no significant improvement for any other variable except for SWE itself, which yields
only positive MSSS values by design. The lack of overall improvement for water-related variables (VOL, SML, GWL, LWL) is coherent with the time delay with changes in SWE. As for the other snow-related variables (STI, SRI), although there may be a relationship with SWE, it is non-linear (U.S. army corps of engineers, 1956), which is further weakened by the distance separating SWE at a snow pillow from STI or SRI at another location.

Only the inclusion of SWE surrounding a given snow pillow in the state vector will be considered during the assimilation of SWE for streamflow predictions.





### 4.1.3 SCA data assimilation

Like streamflow, snow cover area (SCA) is not a state variable. It is computed in parallel with CEQUEAU without having any direct effect on future simulations. In order to have any impact during the assimilation process, there must exist a linear or sufficiently near linear correlation between SCA and state variables. The update step should bring improvements to the

state variables if the computed correlation also reflects the true correlation.

Figure 7 shows a boxplot of the MSSS computed for each variable when they are individually included in the state vector. The open loop scenario is used as a reference. Results show that global snow cover data assimilation yields little or no improvement for all state variables compared to the open loop scenario. For most cases, a strong deterioration is observed,

suggesting that the current use of SCA data is not well adapted for the current assimilation purposes using the EnKF. Marcil et al. (2016) have shown that there exists a relationship between the SCA and the percentage of cumulated streamflow at the outlet, but it is neither linear nor is cumulated streamflow a state variable. The EnKF requirement that relationships between variables be linear and synchronized severely limits the value of global SCA data for the current application. This result is coherent with the findings presented by Clark et al (2006). Using a model which incorporates snow cover area as a state

variable, such as the Snowmelt Runoff Model (SRM; Martinec, 1974) or the Soil and Water Assessment Tool (SWAT; Arnold et al, 1998), could overcome the issue of nonlinearities between variables, while using recursive or smoother approaches to data assimilation could help with the time lag issue between observations and state variables.

Given the absence of overall improvement for all the state variables, the impact of SCA DA on streamflow predictions will

not be considered in this study.

### 4.2 Streamflow forecasts

Aside from granting insight into the sensitivity of the system to the state vector configuration, the analysis in the previous section presented a list of state vector configurations likely to favour streamflow predictions improvements based on the improvement of various state variables. This section will present streamflow prediction results for each configuration

selected for each type of data assimilated.

### 4.2.1 Streamflow data assimilation

Focusing on the case where only streamflow at the outlet are assimilated, Fig. 8 presents the CRPSS of predicted streamflow at the outlet over a forecast horizon of 50 days using the open loop as a reference. Only the state vector configurations that

showed some improvements in the state vector configuration analysis section are shown.





Firstly, high values of CRPSS for short-term forecasts can be observed for the case where only volumes are included in the state vector (blue curve). The CRPSS subsides asymptotically to zero over time, which shows assimilating streamflow to update volumes improves streamflow predictions compared to the open loop only for a few days, after which the impact of streamflow assimilation becomes insignificant. This is not surprising given the nearly linear relationship between

streamflows and volumes. This is coherent with the overall improvement observed for volumes in the state vector configuration analysis (Fig. 4).

Secondly, adding each of the three water reservoirs individually to the volumes yields different results. Even though lake water levels showed improvements over the majority of whole squares, the impact on streamflow predictions (red curve) is

marginal compared to the case where only volumes are included in the state vector. This is because the weights attributed to lakes in CEQUEAU are very low for most whole squares. Only about 0.5% of the entire watershed is modelled using the conceptual lake reservoir and its parameters, unlike the soil moisture and groundwater reservoirs, which are present in every whole squares. Adding SML (green curve) or GWL (magenta curve) instead of LWL increases not only the initial CRPSS, but also slows the decrease over time. This is consistent with the improvements observed for the updated water levels for

over half of the whole squares compared with the open loop case, which translate as added improvements over the case where only volumes are included in the state vector. The slower decrease over time is also coherent with the increase in time it takes for water in the reservoirs to reach the outlet compared with water already in the routing system. The groundwater reservoir is shown to have an initially similar, but longer-lasting positive impact than the soil moisture reservoir. The soil moisture reservoir controls mainly the fast-flowing surface runoff, the amount of evapotranspiration leaving the system and

the amount of water infiltrating into the ground water reservoir. The ground water reservoir has a numerically unlimited capacity, with no way out for the water except through evapotranspiration and the outlets that feed the routing system, making its impact on streamflows last longer than the relatively ephemeral soil moisture reservoir.

Finally, the scenario where all four variables are added to the state vector is analysed (orange curve). The difference noted

with the other curves is mainly caused by the simultaneous inclusion of SML and GWL, since all the other curves already have VOL included, while LWL was already shown to have very little impact on streamflow at the outlet. Comparing the fully combined case with the VOL+GWL case, although the initial improvement is similar between the two, with the former slightly above, the latter has a slower decrease over time. This suggests that the addition of SML interferes with the GWL update. As seen for the state vector configuration analysis (Fig. 4), the assimilation of streamflow at the outlet had a positive

impact on a greater number of whole squares for the GWL than the SML. Here, the increased number of deteriorated SML, which infiltrates into the groundwater reservoirs, hinders the GWL updates such that the results show some deterioration compared with the VOL+GWL case, even though it is still an improvement over the VOL only updates.





### 4.2.2 SWE data assimilation

Following the same method as with streamflow, this section focuses on the case where only SWE from snow pillows are assimilated. Since the state vector configuration analysis showed only improvement for the SWE variable, it will be the only variable added to the state vector for streamflow forecast. However, since there are three observations available, Fig. 9 presents the CRPSS of predicted streamflow at the outlet when assimilating SWE from each snow pillow individually and collectively.

An interesting result is that the impact of each snow pillow on streamflow predictions varies greatly. The impact of the snow pillows located at Mount Wells (green curve) and Mount Pondosy (blue curve) are dwarfed in comparison with the impact of the snow pillow located at Tahtsa Lake (magenta curve). This is coherent with results from Marcil et al (2016) over the same watershed. The lower impact of the Mount Pondosy snow pillow is explained by the relatively small region of influence observed in Fig. 5b. As for Mount Well, even though it has the largest area of influence (Fig. 5a), it is also the snow pillow affecting the regions with the lowest altitudes and also the least amount of maximum SWE. Although the regions affected by the Mount Wells snow pillow contain a mean annual maximum SWE of 410 mm, it is 40% less than for the region affected by the Tahtsa Lake snow pillow, which sits at 682 mm.

Nonetheless, assimilating all three snow pillows yields better results for mid-term streamflow forecasts (Fig. 9, red curve). Even though the Tahtsa Lake snow pillow carries the most importance, the other snow pillows have a positive effect on regions that are not reached by the Tahtsa Lake snow pillow area of effect. The assimilation of all three snow pillows does yield short-term forecasts improvements, but a better score is reached over time. This is because the impact of SWE over streamflow at the outlet occurs during snowmelt, which can occur at a much later date than when SWE observations are assimilated. Although the curve gives the impression of a monotonous increase over time, this is only due to the limit imposed on the forecast horizon. At a further horizon, the curve should eventually peak and decrease asymptotically to zero since there is no accumulation of snow from year to year. Over time, the simulation should eventually become indistinguishable from the open loop scenario.

### 4.2.3 Combined streamflow and SWE data assimilation

The focus now shifts to the case where streamflow at the outlet are simultaneously assimilated along with SWE from the three snow pillows. The state vector configuration which provided the best results from the streamflow data assimilation case are used (VOL+GWL) along with the best configuration from SWE data assimilation (SWE only). Although these configurations worked best with their respective data assimilation case, they could behave differently when both streamflow and SWE are assimilated together.





Table 1 presents four configurations for the combined assimilation of streamflow and SWE observations. These configurations differ in the overlap of their effect during the update phase such that some configurations allow both observations to simultaneously update the same variable, while others do not.

The performance of these configurations on the CRPSS for predicted streamflow at the outlet is presented in Fig. 10. While all four configurations perform in a very similar way for short-term streamflow predictions, the group forms two pairs that differ in that the blue-magenta group allows SWE observations to update modelled streamflow, while the green-red pair does not. Although allowing SWE data assimilation to update VOL and GWL changes very little, a drop in performance occurs if streamflow assimilation updates modelled SWE. This is coherent with the state vector configuration analysis performed in

the previous section (Fig. 4 and Fig. 6), where SWE data assimilation is shown to have a weak impact on VOL and a median MSSS around 0 for GWL, while streamflow assimilation deteriorated around 75% of SWE whole squares when they were included in the state vector.

Overall, the simultaneous assimilation of streamflow and observed SWE yields important improvements over the entire

forecast horizon analysed, with the streamflow data assimilation improving mainly short-term streamflow forecasts and SWE data assimilation improving mainly mid-term streamflow forecasts.

Furthermore, the two data types differ not only by their impact over the forecast horizon, but also over the time of the year. Fig. 11 and Fig. 12 show the monthly CRPS for the open loop (black curve), the streamflow data assimilation including

VOL and GWL (blue curve), the SWE data assimilation of all snow pillows (red curve) and the simultaneous, but separated, streamflow and SWE data assimilation (VG-S; green curve) for short-term (average of horizon from 1 to 5 days) and mid-term (average of horizon from 25 to 50 days) streamflow forecasts. The CRPS is shown for the open loop to show the performance change over the time of the year and the period when improvement is most needed. Recall that the CRPS ranges from zero to infinity, with zero being a perfect forecast. The period from May to July, which corresponds to the melt

period, is therefore the period when the CRPS is highest for short- and mid-term forecasts are the most problematic. The scores for mid-term forecasts are lower than for short-term forecasts because the true weather is used as input for forecasts such that the open loop slowly converges to the true states over time.

Assimilating streamflow results in an improved score over the entire year for short-term forecasts, although little gain is

obtained for mid-term forecasts. This steady improvement is to be expected since streamflow here is always nonzero and observations are available all-year round. On the other hand, the impact of the assimilation SWE from snow pillows is limited mainly to the melt period for both short-term and mid-term forecasts. However, this period corresponds to the problematic period when most gain can be obtained. The assimilation of SWE provides a better score than the assimilation of streamflow for the same period and improves going from short-term to mid-term forecasts. SWE assimilation complements



streamflow assimilation as observed from the performance of the simultaneous assimilation, which yields both the steady improvements over the year and the important gain during the snowmelt period.

# 5 Conclusion

This study investigated the impact that multivariate data assimilation can have on streamflow forecasts using the CEQUEAU hydrologic model applied over the Nechako watershed in a synthetic experiment. The study also showed the importance of the state vector configuration on streamflow forecasts when using the EnKF.

Streamflow data assimilation was found to improve short-term streamflow forecast considerably. However, the impact
dissipated relatively rapidly as a function of the forecast horizon, which was slowed by adding groundwater conceptual reservoir levels to the state vector. Improvements were observed for all months of the year; low-flow and high-flow periods alike.

On the other hand, the assimilation of snow water equivalent data from synthetic snow pillow data yielded streamflow
forecast improvements mainly during the snowmelt period. Although the period lasts approximately three months, the impact was found to be greater than streamflow data assimilation over the same period. It was also noted that assimilating each snow pillow data individually yielded different results, with various radii of influence, such that the improvement from assimilating all three snow pillows simultaneously covered most of the watershed and yielded streamflow forecasts which outperformed forecasts from any single snow pillow data assimilation. Over the forecast horizon, the peak of improvement
was greater than or equal to the 50 days limit over which forecasts were simulated, which contrasts the short-lived impact of streamflow data assimilation.

Given their complementarity, streamflow and snow water equivalent data were assimilated simultaneously. The resulting streamflow forecast inherited the strengths from both types of data, having a strong, positive impact for both short-term and
mid-term forecasts. Improvements were obtained for all periods of the year, but mainly during the snowmelt period, which is normally the most problematic.

The assimilation of basin-wide snow cover area failed to improve the simulation of any state variable, which deteriorated streamflow forecasts. The most probable factor was determined to be the absence of snow cover area as a state variable or a
proxy with a sufficiently linear relationship with SCA. This was compounded by the nonlinear relationship between SCA and the state variables, which is required by the EnKF. Suggestions to improve the method to accommodate for snow cover





area are to use a model which incorporates snow cover area as a state variable and/or to use a data assimilation approach which takes into account a time lag between observations and state variables.

The results obtained are conditional to some hypotheses. First, since this is a synthetic experiment, it is assumed that a real
5   experiment would behave similarly to a simulation using CEQUEAU with a specific set of parameters and inputs. Second, the potential impact of data assimilation on streamflow forecasts observed depended on using the true weather inputs. Using real weather inputs may decrease this impact. Third, it is assumed that the error representations for the model inputs and the observations are known, independent and unbiased. Finally, the impact of errors from the model parameters is assumed to be negligible, such that the set of parameters was not altered from the true simulation's set of parameters.

The hypotheses listed reveal several challenges posed by the assimilation of multiple types of observations for streamflow forecasting purposes. Future works include investigating into the performance of multivariate data assimilation in the presence of biases and unknown errors, as well as the economic impact of streamflow forecasts generated with multivariate data assimilation on real management practices.

*Acknowledgements*. We thank Rio Tinto for their collaboration in this study, providing funding, data and the hydrologic model CEQUEAU. This research is also funded by Quebec's and Canada's funding agencies FRQNT and NSERC, respectively.

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



**Table 1: Overview of multivariate DA scenarios.**

| Method | Streamflow DA updates: | | SWE DA updates: | |
|---|---|---|---|---|
| | VOL + GWL? | SWE? | VOL + GWL? | SWE? |
| VG-S | yes | no | no | yes |
| VG2-S | yes | yes | no | yes |
| VG-S2 | yes | no | yes | yes |
| VG2-S2 | yes | yes | yes | yes |



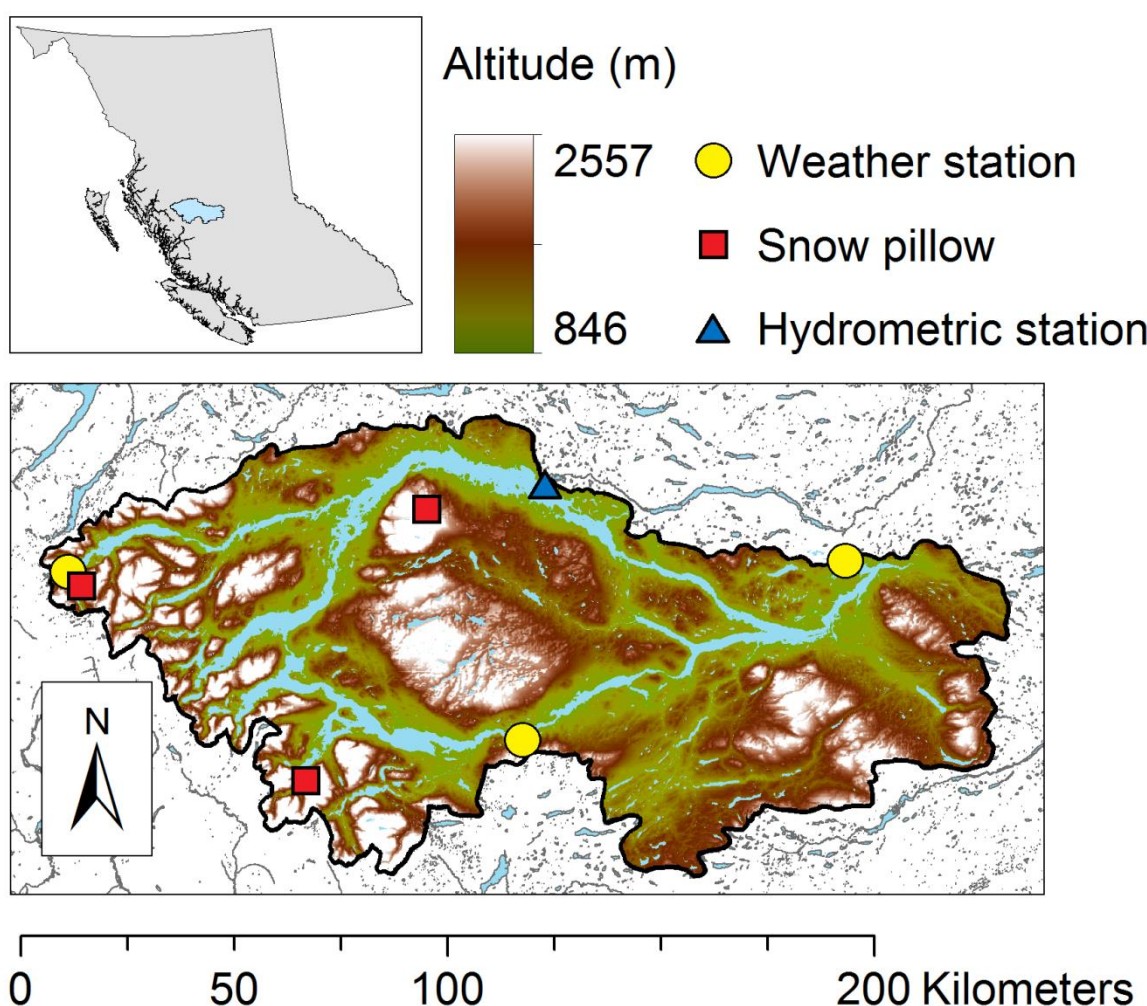

**Figure 1: The Nechako watershed and the locations of weather stations, snow pillows and a hydrometric station. All of these contain at least daily weather data. The outlet is considered to be at the spillway, located at the blue triangle. The intake is located at the Tahtsa Intake weather station (westernmost yellow circle).**





**Figure 2: Diagram of the processes included in CEQUEAU's production function.**




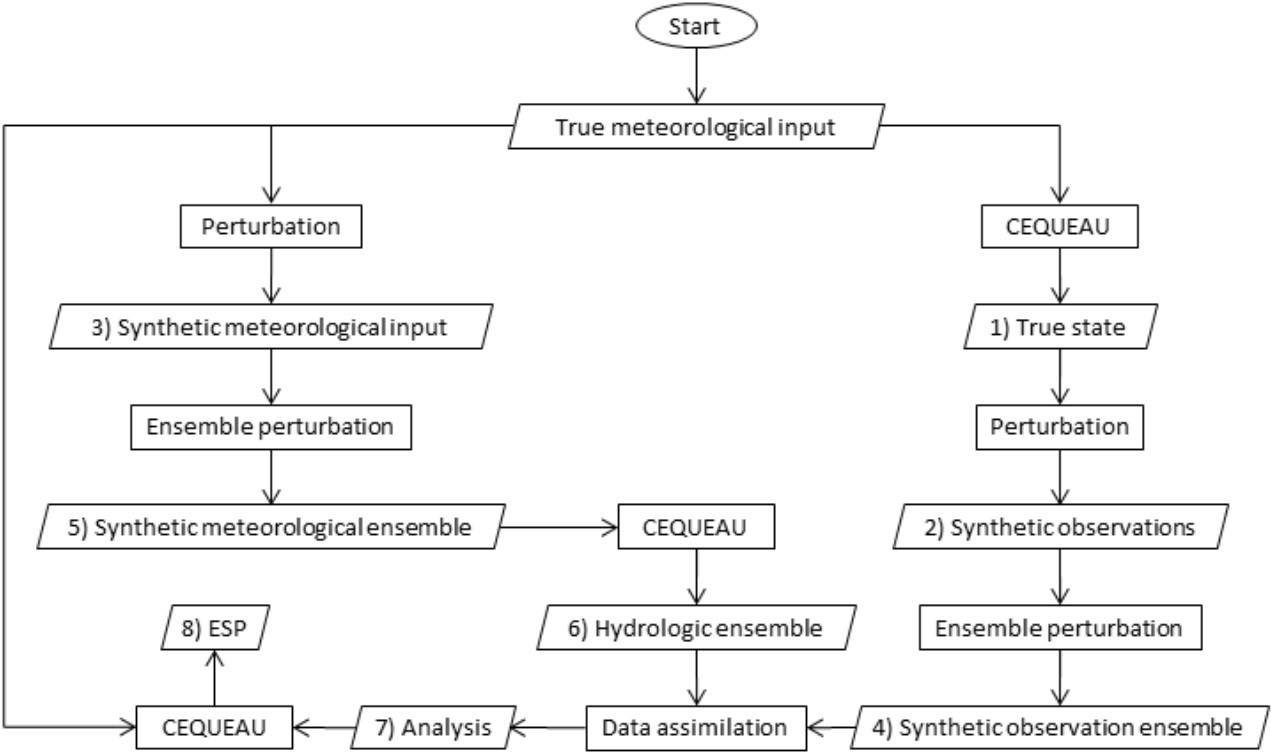

**Figure 3: Flowchart for the production of ensemble streamflow predictions obtained from various data assimilation scenarios.**




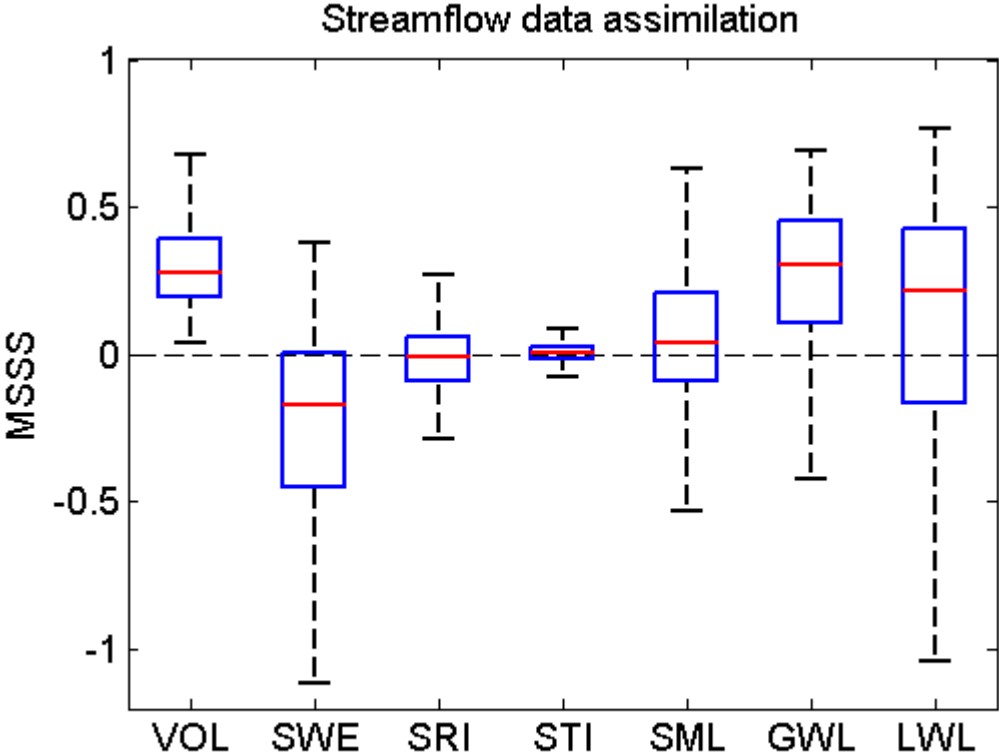

**Figure 4: Box plot of the Mean Square Skill Score for each variable when assimilating streamflow at the outlet. The open loop is used as a reference. Outliers are not shown for visibility purposes.**





**Figure 5: Distribution of the Mean Square Skill Score of SWE over the watershed when assimilating SWE located at a) Mount Wells, b) Mount Pondosy and c) Tahtsa Lake. The open loop is used as a reference. Values below -1 are cut off from the legend.**





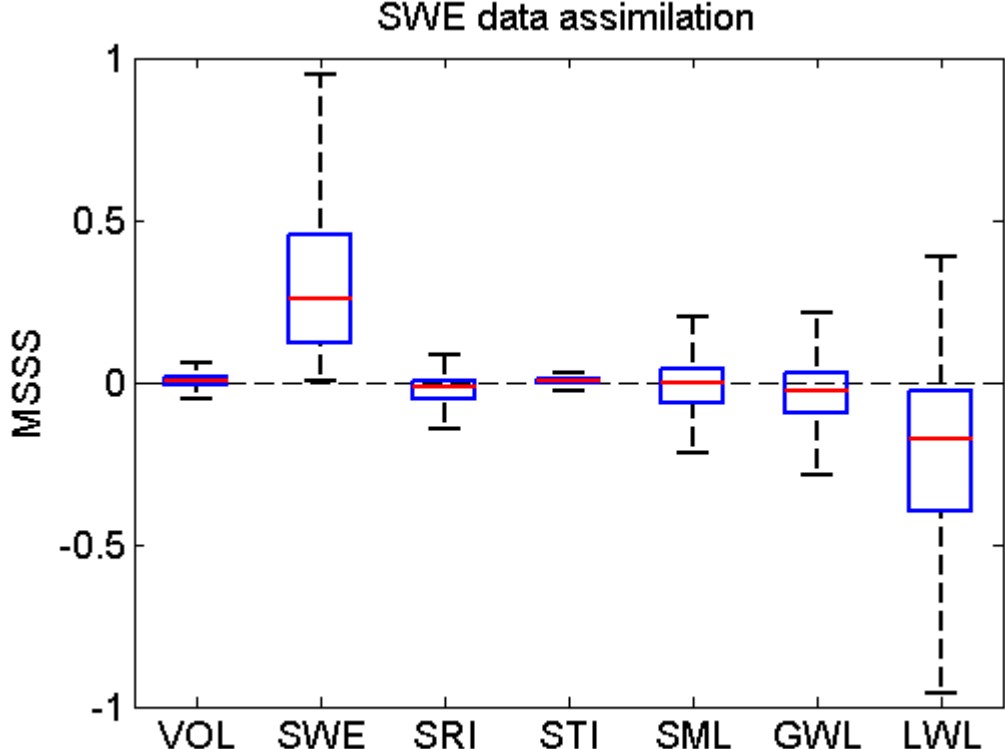

**Figure 6: Box plot of the Mean Square Skill Score for each variable when assimilating SWE from all three snow pillow locations. The open loop is used as a reference. Outliers are not shown for visibility purposes.**




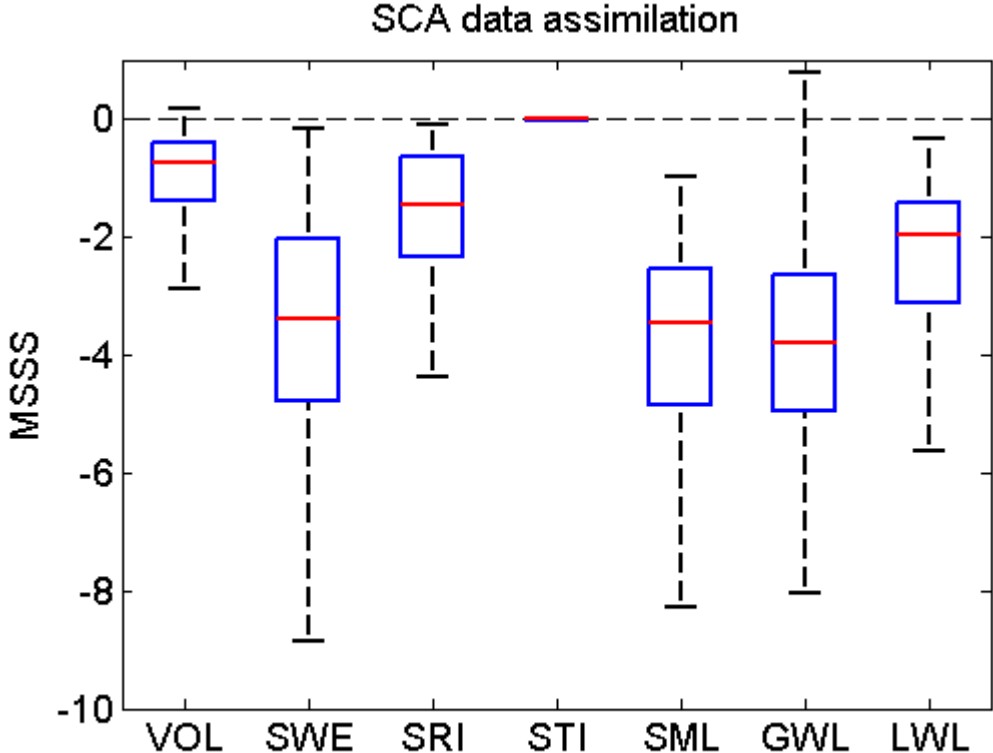

**Figure 7: Box plot of the Mean Square Skill Score for each variable when assimilating basin-wide snow cover area. The open loop is used as a reference. Outliers are not shown for visibility purposes.**




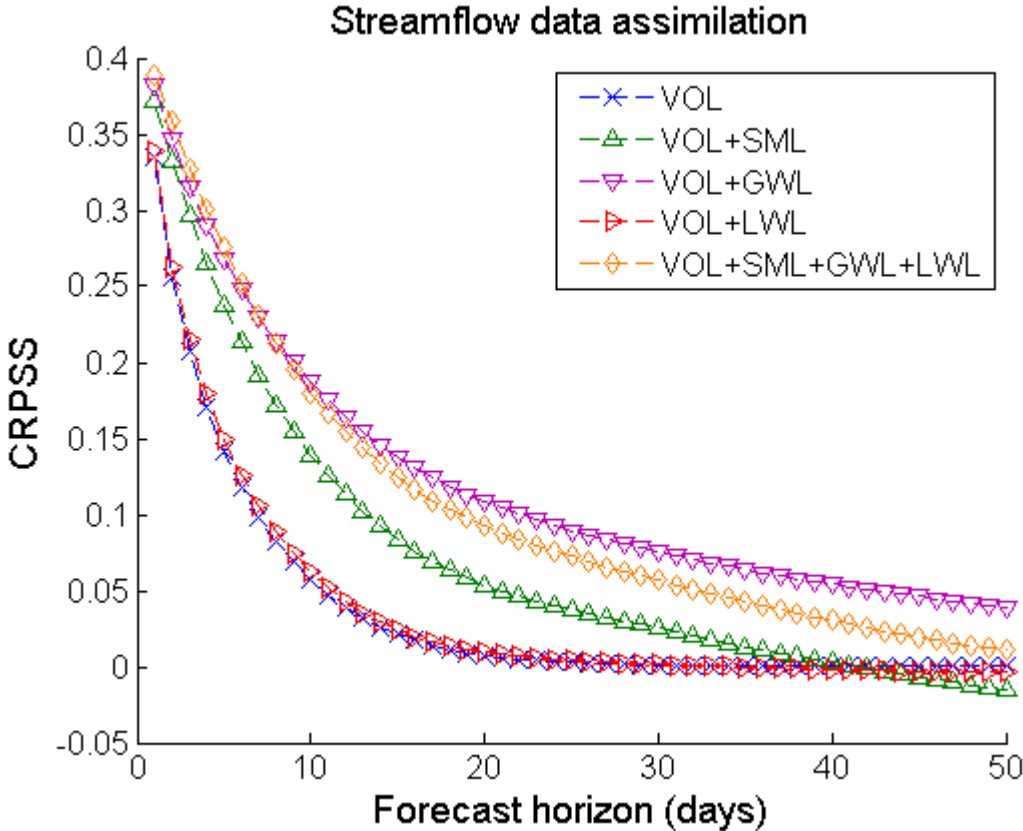

**Figure 8: Continuous Rank Probability Skill Score of the streamflow ensemble when assimilating streamflow at the outlet. The open loop is used as a reference. The forecast horizon varies from 1 to 50 days.**





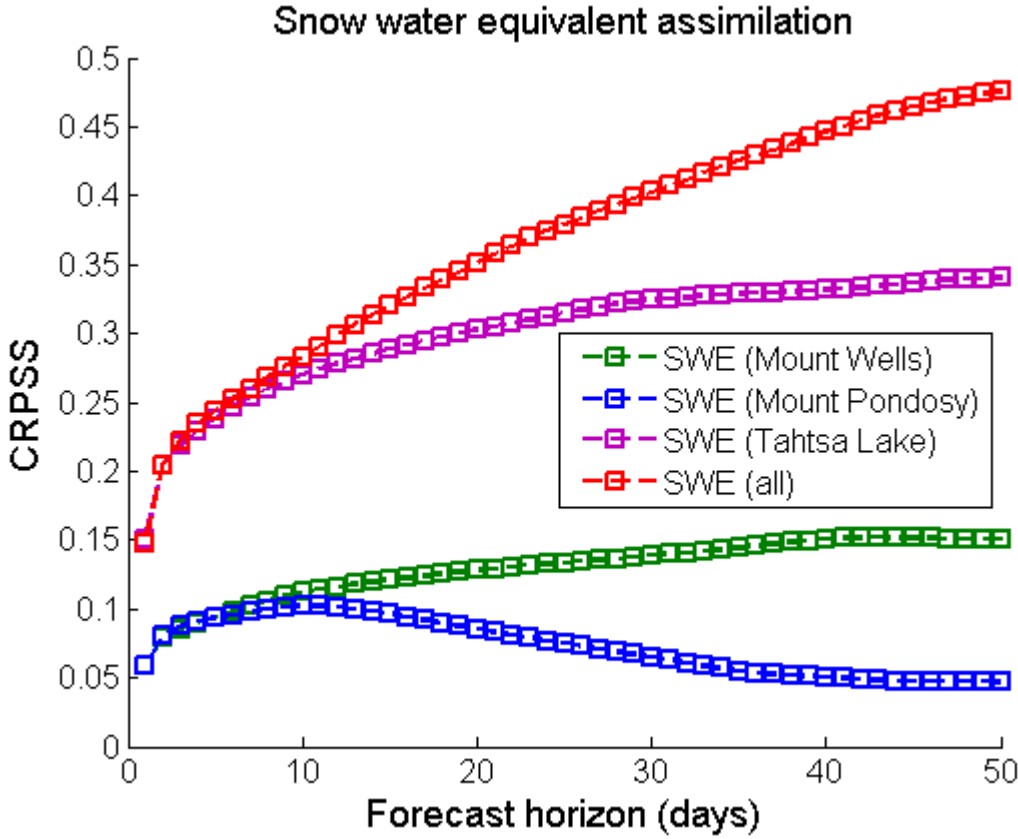

**Figure 9: Continuous Rank Probability Skill Score of the streamflow ensemble when assimilating SWE from all three snow pillow locations. The open loop is used as a reference. The forecast horizon varies from 1 to 50 days.**





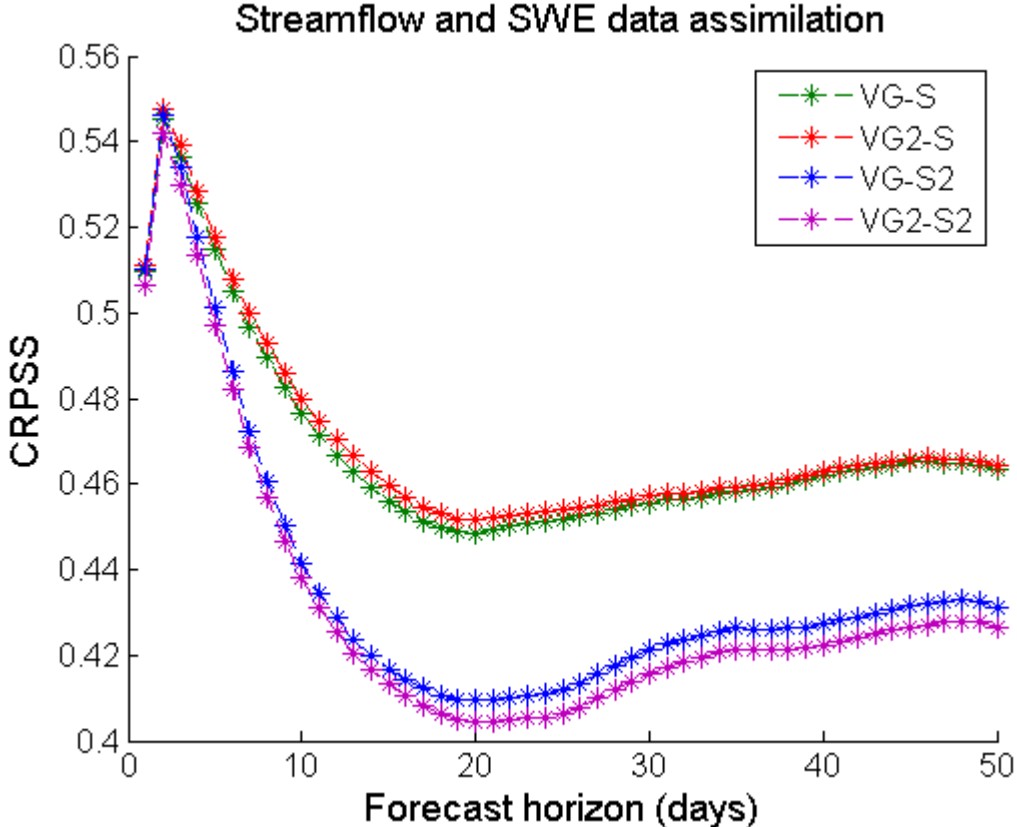

**Figure 10: Continuous Rank Probability Skill Score of the streamflow ensemble when assimilating streamflow at the outlet and SWE from all three snow pillow locations. The open loop is used as a reference. The forecast horizon varies from 1 to 50 days. Lack of parentheses indicates that the variable is affected by both types of observations.**




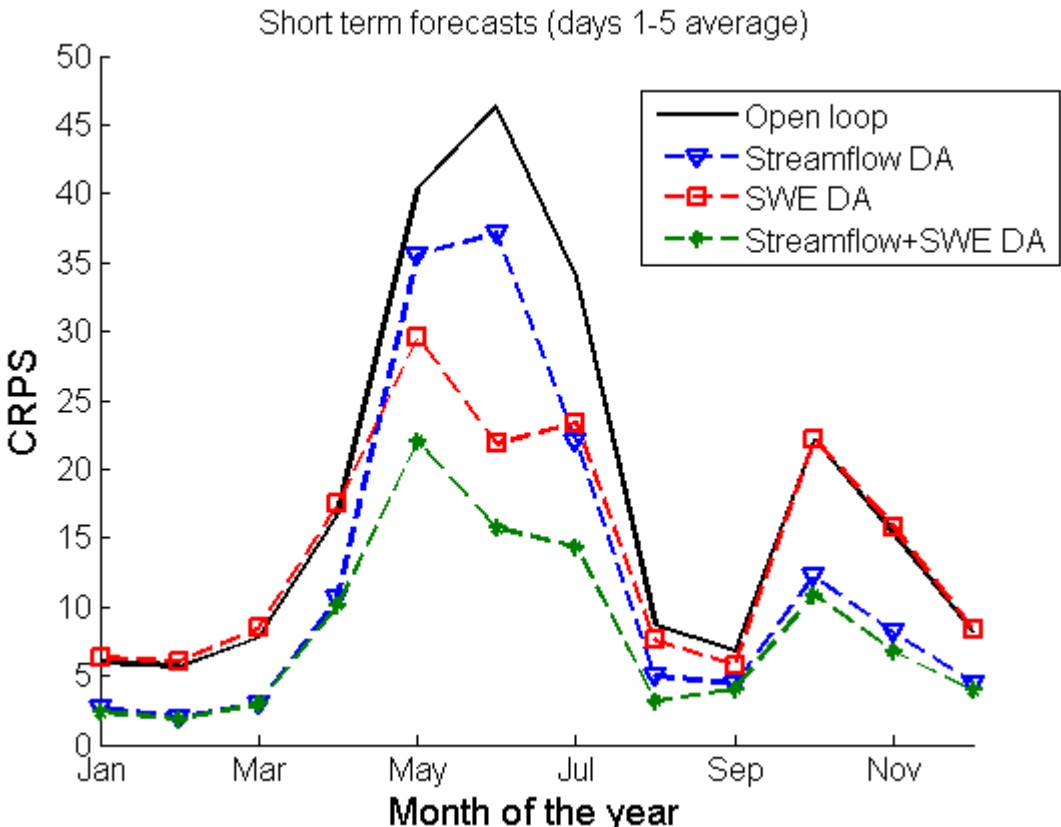

**Figure 11: Continuous Rank Probability Score for short-term forecasts (average of forecast horizons 1 through 5 days) of various data assimilation scenarios as a function of the month of the year.**





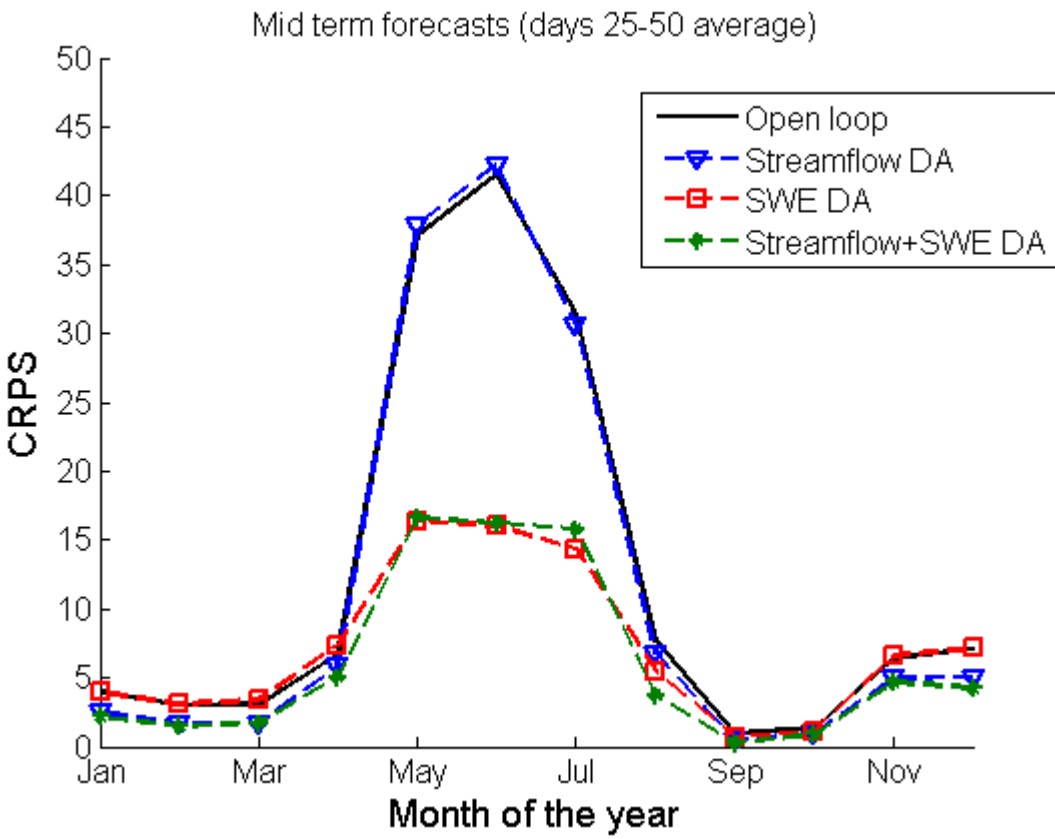

**Figure 12: Continuous Rank Probability Score for mid-term forecasts (average of forecast horizons 25 through 50 days) of various data assimilation scenarios as a function of the month of the year.**

