# Peer review of "Combined assimilation of streamflow and snow water equivalent for mid-term ensemble streamflow forecasts in snow-dominated regions"

_Hydrology and Earth System Sciences, 2016_

## Referee Comment (RC1) · Anonymous Referee #1 · 1 Jun 2016

In this study, the authors explore the assimilation of discharge, SWE and SCA in a hydrologic model for the potential to improve streamflow forecasting in a mountainous basin in western Canada. Synthetic data sets are developed and used. The authors first determine which state variables are adequately predicted by the three data types that are candidate for assimilation. SCA was found to not be a good predictor. Then, the impact of assimilating SWE and discharge on hydrologic forecasts was tested.

Overall, this is an interesting study with good results. Forecasts were improved with SWE and Q assimilation both when assimilated individually and simultaneously. It is demonstrated that the data were useful for adjusting several model states (VOL, GWL, and SWE) in the CEQUEAU model. The methods in this paper show promise

for applications in forecasting provided the results remain consistent for non-synthetic studies.

General comments:

1) There needs to be more detail provided in the methods section. As is, it appears as though the methods are valid, but I could not replicate this study with the information provided. I find myself having to assume I know what the authors did during some steps. Therefore, specific comments about where to add necessary detail are provided below.

2) In the results section, the authors should make a stronger effort to link their findings to other studies. There are several papers referenced that explore assimilation of SWE and/or discharge in snow-dominated areas. There are also likely studies that have examined this type of data in other modeling and forecasting contexts. While there are a few comments about results from other studies, the authors should try to add more to the discussion.

Specific comments: Page 2, line 30. The last sentence might be better as a statement rather than a question. It seems out of character with the rest of the introduction.

Page 3, line 9: "such that the difference in elevation reaches about 1700m" is oddly stated. The difference in elevation between what? If 1700m is the total relief of the mountain, simply state it that way.

Page 4, line 8: US Army Corps of Engineers should be capitalized. Also, is the inclusion of "1956" intended to be a citation? There is nothing listed in the references regarding this.

Page 4, line 17: SI is not in equation 1. How is it relevant to this discussion?

Page 4, Line 18-19: As written it is implied that Hall et al. (2002) calibrated the three parameters in Equation 1. I do not think that is the case. Additional explanation of this calibration is needed. Who conducted the calibration? Was it conducted for this

region? If not, is it considered to be universally applicable?

Page 4, line 20: Tampered not tempered.

Page 6 , line 6-8: this statement is becoming repetitive. It was mentioned several times in this section that it has been shown useful in hydrologic studies. I recommend removing earlier statements like this, or combine them into one or two sentences.

Page 6, lines 14-24: It isn't clear what variables are referred to when using the term "observations". This may be stated earlier, but it would help the reader if they were explicitly stated here. In general, this section lacks detail. In what way and by how much were the data perturbed? How do you get synthetic observations by perturbing "true states"? More specific terminology and combining or pulling in information from Section 3.2 would be helpful in understanding the procedure of creating synthetic data.

The methods section includes very little description of how ESP forecasts are generated. A more thorough explanation should be provided for readers unfamiliar with the process. Please clarify whether only 20 years of meteorological data (1990-2000) were used to generate the ESP ensembles. Also, was only the mean value of the state variable predicted by the EnKF used to generate each ensemble in the ESP forecast, or were multiple state values from the state variable ensemble used?

Page 10, line 24 onward: What is the timestep of the data evaluated? Hourly, daily, etc? I cannot find where this is explicitly stated, but it is important to understanding the results of the study. If the streamflow is evaluated at a daily timestep, it makes sense that the SWE is not beneficial for predicting streamflow; however, results might be quite different if output is evaluated at a monthly or seasonal timestep. In addition, I could not find the interval between assimilation, is it done at each model timestep, daily, weekly?

Page 14, lines 1-6 and Figure 8: It is not quite clear what VOL is in the model. As presented on page 4, it appears to be water that is being routed to the outlet (i.e.

runoff). If that is the case, the quick decline in adjusting the VOL state on the CRPSS makes sense not because of a linear relationship with discharge, but because of the likely short residence time of the water represented by VOL within the watershed. The authors discuss the residence time issue in the next paragraph with respect to GWL and SML, I would like to see similar insight regarding the VOL as the linear relationship explanation is not obvious.

Page 16, lines 14-16 and Figure 10: It would be helpful to add a sentence putting results from Figure 10 in context of results from assimilating only Q or only SWE.

Page 17, lines 28-29: The SCA was not tested on the forecasts due to the lack of improvements in state variables (page 13, lines 19-20). The authors should not make any conclusions regarding the impact on forecast skill from this study, and restate this with respect to state variable improvements.

Page 18, line 4: The statements made in this paragraph are not hypotheses, they are assumptions and limitations.

---

## Referee Comment (RC2) · Anonymous Referee #2 · 20 Jun 2016

The manuscript titled Combined assimilation of stream flow and snow water equivalent for mid-term ensemble stream flow forecasts in snow-dominated regions is an attempt to apply the Ensemble Kalman Filter to the application of stream flow prediction in regions where the majority of the water involved comes from snow.

The manuscript undertakes a set of studies, the first is to ascertain which of the 7 possible state variables they consider are sensitive to the assimilation of the 3 observation types they consider. They indicate that snow cover appears to not be an important factor in the forecasts that they seek.

The manuscript is well written and upon a second reading easy to follow. However, i do have a couple of points that need to be addressed before i can sign off on publication.

Major Comments: 1) My first concern relates to the generation of the ensemble perturbations and the perturbations to the observations. You indicate that you use Gamma, lognormal of beta distributions yet the EnKF is highly reliant on these perturbations, and hence the errors being Gaussian distributed. My query, and i am requesting graphs of these, is to see the distributions plots for the distributions that you mentions with the parameters in the manuscript. My hunch is that these will look quite close to a Gaussian distribution of some form and as such is why you obtain the results, which are great results, but it could be misleading to have these distributions when really they are close to a Gaussian.

2) You need to provide a better justification to the use of these distributions on page 7.

3) You need to rewrite the paragraph starting on page 6 at line 14 as it is confusing as it would appear that it looks like you are referring to equations.

4) The statement on page 17, line 30 does not make sense and is confusing about the need for linear relationships which you really should have with the EnKF.

5) On page 9 you are finishing the details about the localization but i am concerned that because you achieve this wrt the true state that this may not be the case in the real data situation and you need some sort of disclaimer here as you are kind of using the true localization which would not be the case in reality.

Minor comments: 1) Page 3, line 27 remove to 2) Page 5, line 17, you mention the gain yet you have not defined it. 3) Page 11, line 33, remove the first that 4) Page 15, line 15, sits is not a very scientific way to describe where the site is. 5) General comment. you use both Gaussian and normal please be consistent and only use one of them

---

## Referee Comment (RC3) · K. He (Referee) · 7 Jul 2016

The authors present a set of synthetic experiments in assessing the potential added value of assimilating streamflow, SWE, SCA (via EnKF) into the CEQUEAU model in short- to medium-range streamflow forecasting at the Nechako watershed located in BC, Canada. Results indicate that streamflow DA and SWE DA lead to improvements in short-term forecast and medium-term forecast (during snow melt period), respectively. Assimilation of streamflow and SWE simultaneously yields even better results at both scales. However, assimilating SCA does not show any benefit. Overall, the methodology and results are sound and meaningful, yet not innovative. The paper is very well written and organized. I think it will be of interest to the readership of HESS.

My major comment is that, from the perspective of water resources management, the bias of the mean (or median) ESP forecast is typically an important factor considered in water-related decision making (e.g., water supply allocation, reservoir release/hydro-power generation schedule, among others). In light of this, when assessing EPS forecast skill, the bias is normally analyzed. In the case of this study, the score MSSS is applied in the sensitivity analysis part (Figures 4, 6 and 7) but not the forecast part (Figures 8-12). The relevant results should be added (either in tabular or graphic form) and discussed.

My minor comments include 1) the authors need to be clear about how often the forecasts are issued (every day, once a week, or once per month in the study period from 8/15/1990 to 8/14/2000). If it is once a month, the authors need to discuss the sample size issue (10 years) when discussing the skill scores; 2) Line 7 of Page 2, "Franz" should be "Franz et al."; 3) Lines 26-27 of Page 3, august should be August; delete "to"; 4) Line 8 of Page 4, (Fig. 2), Army Corps of Engineers; 5) Lines 14-23 of Page 6, other than use (1), (2),..., it would less confusing when using (#1), (#2), ..., or (Step 1), (Step 2), etc. ; 6) Line 5 of Page 7, delete "to"; 7) Line 6 of Page 8, modify "than"; 8) Lines 20-22 of Page 8, rework on the sentence; 9) Line 12 of Page 10, change MSS to MSSS; 10) Line 20 of Page 10, in order to.

---

## Author Comment (AC1) · 26 Jul 2016

Italicized text: Reviewer's comment

AR: Authors' response

**Comments to reviewer #1**

*In this study, the authors explore the assimilation of discharge, SWE and SCA in a hydrologic model for the potential to improve streamflow forecasting in a mountainous basin in western Canada. Synthetic data sets are developed and used. The authors first determine which state variables are adequately predicted by the three data types that are candidate for assimilation. SCA was found to not be a good predictor. Then, the impact of assimilating SWE and discharge on hydrologic forecasts was tested.*

*Overall, this is an interesting study with good results. Forecasts were improved with SWE and Q assimilation both when assimilated individually and simultaneously. It is demonstrated that the data were useful for adjusting several model states (VOL, GWL, and SWE) in the CEQUEAU model. The methods in this paper show promise for applications in forecasting provided the results remain consistent for non-synthetic studies.*

*General comments:*

*1) There needs to be more detail provided in the methods section. As is, it appears as though the methods are valid, but I could not replicate this study with the information provided. I find myself having to assume I know what the authors did during some steps. Therefore, specific comments about where to add necessary detail are provided below.*

AR: Information has been rearranged into new sections and additional details were added as recommended in the specific comments section. Section 3.1, dealing with the overall approach used during the synthetic experiment, now has three subsections explaining specifically how synthetic observations (3.1.1), synthetic meteorological input (3.1.2) and ESPs (3.1.3) were generated. Sections 3.2.2 and 3.2.3 now deal exclusively with meteorological ensemble and observation ensemble generation, respectively. Some information may appear to be redundant at first, but this is because similar approaches were used to generate perturbations required in the creation of single-valued synthetic observations/meteorological input and their ensemble version.

*2) In the results section, the authors should make a stronger effort to link their findings to other studies. There are several papers referenced that explore assimilation of SWE and/or discharge in snow-dominated areas. There are also likely studies that have examined this type of data in other modeling and forecasting contexts. While there are a few comments about results from other studies, the authors should try to add more to the discussion.*

AR: Comparisons with other studies have been added in multiple sections, including the state vector configuration when assimilating streamflow (4.1.1) and the impact of streamflow (4.2.1) and SWE (4.2.2)

data assimilation on streamflow forecasts. Additional references will also be introduced for comparison purposes.

*Specific comments:*

*Page 2, line 30. The last sentence might be better as a statement rather than a question. It seems out of character with the rest of the introduction.*

AR: The sentence has been reformulated as a statement:

"The importance of state vector configuration when using multivariate DA for hydrological modeling has yet to be investigated."

*Page 3, line 9: "such that the difference in elevation reaches about 1700m" is oddly stated. The difference in elevation between what? If 1700m is the total relief of the mountain, simply state it that way.*

AR: The difference in elevation between the highest and lowest point in the watershed reaches about 1700m. This has been clarified in the revised manuscript.

*Page 4, line 8: US Army Corps of Engineers should be capitalized. Also, is the inclusion of "1956" intended to be a citation? There is nothing listed in the references regarding this.*

AR: The missing reference has been added and capitalized.

*Page 4, line 17: SI is not in equation 1. How is it relevant to this discussion?*

AR: SI is in the denominator of equation 1. It is one of three parameters used to convert snow water equivalent into snow cover area. It is included, along with other parameters, to be transparent in our approach and help readers understand the method used to produce the results shown in the manuscript.

*Page 4, Line 18-19: As written it is implied that Hall et al. (2002) calibrated the three parameters in Equation 1. I do not think that is the case. Additional explanation of this calibration is needed. Who conducted the calibration? Was it conducted for this region? If not, is it considered to be universally applicable?*

AR: The reference was initially meant to be a reference for the MODIS data, but this was not clear from the way the sentence was structured.

All the model parameters were not calibrated in the same way. CEQUEAU has numerous parameters that have been manually calibrated by engineers working for Rio Tinto, our industrial partner. These include snow, soil, evapotranspiration and transfer parameters. However, since snow cover area is not explicitly computed by CEQUEAU, a depletion curve had to be appended. The details pertaining to the depletion is not found in CEQUEAU's user manual or in any other study using or detailing how CEQUEAU works. To be transparent in our approach, we explain how the depletion curve is computed, requiring three parameters. The calibration of those parameters was conducted using the SCE-UA method (Duan et al, 1992) to minimize the root mean square difference between simulated snow cover area and MODIS data for each whole square within the Nechako watershed. The parameters are therefore not likely to be "universally applicable", but the approach may be.

The sentence has been reworked in the revised version of the manuscript to include more information about the calibration process.

*Page 4, line 20: Tampered not tempered.*

AR: This has been corrected in the revised manuscript for all occurrences of the term.

*Page 6, line 6-8: this statement is becoming repetitive. It was mentioned several times in this section that it has been shown useful in hydrologic studies. I recommend removing earlier statements like this, or combine them into one or two sentences.*

AR: The sentence has been modified to better transition into the following section.

*Page 6, lines 14-24: It isn't clear what variables are referred to when using the term "observations". This may be stated earlier, but it would help the reader if they were explicitly stated here.*

AR: The term "observations" used in the manuscript refers to observations to be assimilated using the EnKF, namely streamflow, SWE and SCA. The expression "meteorological input" or "weather input" is used to refer to precipitation and mean air temperature. There is one occasion in the manuscript where the expression "meteorological observation" is used and has been corrected to avoid confusion. Additional clarifications have also been added to explicitly state what are "observations" and "meteorological input".

*In general, this section lacks detail. In what way and by how much were the data perturbed? How do you get synthetic observations by perturbing "true states"? More specific terminology and combining or pulling in information from Section 3.2 would be helpful in understanding the procedure of creating synthetic data.*

AR: Initially, sections 3.2.2 and 3.2.3 described how both the individually perturbed and ensemble version of meteorological input and observations were generated since they both used a similar approach using the same perturbation factors. The information was added in the "Hyper-parameter tuning" section since it partly dealt with specifying errors, which is required when generating ensembles.

However, this could lead to confusion between the two sets and appeared to create a void in the description of the synthetic experiment. To avoid confusion, the information from sections 3.2.2 and 3.2.3 strictly dealing with initial perturbations of observations and meteorological input have been used to create additional subsections (3.1.1 and 3.1.2) following the overview of the synthetic experiment.

*The methods section includes very little description of how ESP forecasts are generated. A more thorough explanation should be provided for readers unfamiliar with the process. Please clarify whether only 20 years of meteorological data (1990-2000) were used to generate the ESP ensembles. Also, was only the mean value of the state variable predicted by the EnKF used to generate each ensemble in the ESP forecast, or were multiple state values from the state variable ensemble used?*

AR: ESPs were generated everyday over the entire study period (10 years) using a forecast horizon spanning 50 days. The multiple state values resulting from the EnKF were used to generate ESPs and is the only factor differentiating ensemble members during the forecast phase. The same true meteorological input and model parameters were during the forecast phase. No meteorological ensemble was used to generate ESPs. Although this generates forecasts which are not "realistic" since they get better over the forecast horizon, this is a way to evaluate the potential impact of data assimilation on streamflow forecasts, which is one of the two goals of the study. Using a meteorological ensemble forecast would simply add unnecessary noise. An additional section (3.1.3) has been added on ESP generation.

*Page 10, line 24 onward: What is the timestep of the data evaluated? Hourly, daily, etc? I cannot find where this is explicitly stated, but it is important to understanding the results of the study. If the streamflow is evaluated at a daily timestep, it makes sense that the SWE is not beneficial for predicting streamflow; however, results might be quite different if output is evaluated at a monthly or seasonal timestep. In addition, I could not find the interval between assimilation, is it done at each model timestep, daily, weekly?*

AR: The time step is daily. This has been clarified in the revised manuscript by mentioning a daily time step for the model and daily availability for the observations.

*Page 14, lines 1-6 and Figure 8: It is not quite clear what VOL is in the model. As presented on page 4, it appears to be water that is being routed to the outlet (i.e. runoff). If that is the case, the quick decline in adjusting the VOL state on the CRPSS makes sense not because of a linear relationship with discharge,*

*but because of the likely short residence time of the water represented by VOL within the watershed. The authors discuss the residence time issue in the next paragraph with respect to GWL and SML, I would like to see similar insight regarding the VOL as the linear relationship explanation is not obvious.*

AR: The initial impact strength is due to the close link between streamflow and VOL. Since the correlation between the streamflow observation and VOL is relatively high, VOL globally experiences a positive update (as seen in figure 4). In return, since simulated streamflow depends strongly on VOL, simulated streamflow initially experiences an important positive impact. However, the duration of this impact is indeed caused by the water residence time. A short-lived impact on CRPSS as a function of forecast horizon would be caused by a relatively short residence time. This was not obvious from the way it was stated in the initial manuscript and has been clarified. Additional clarifications have also been added to section 2.2 concerning the role of VOL and streamflow.

*Page 16, lines 14-16 and Figure 10: It would be helpful to add a sentence putting results from Figure 10 in context of results from assimilating only Q or only SWE.*

AR: A few sentences have been added to address the comparison between the combined and individual data assimilation scenarios:

"CRPSS values for combined assimilation of both streamflow and SWE observations were superior to CRPSS values for individual assimilation of streamflow or SWE over all forecast horizons, with the exception of forecast horizons higher than 45 days, where CRPSS values for SWE DA are slightly higher. This reveals that while the updated VOL and GWL by streamflow data assimilation may be very beneficial for short-term forecasts, they do not further improve the mid-term forecasts when combined with SWE data assimilation in comparison with the scenario where only SWE data is assimilated."

*Page 17, lines 28-29: The SCA was not tested on the forecasts due to the lack of improvements in state variables (page 13, lines 19-20). The authors should not make any conclusions regarding the impact on forecast skill from this study, and restate this with respect to state variable improvements.*

AR: This has been corrected in the revised manuscript.

*Page 18, line 4: The statements made in this paragraph are not hypotheses, they are assumptions and limitations*

AR: This has been corrected in the revised manuscript.

References :

Duan, Q. Y., Sorooshian, S. and Gupta, V.: Effective and Efficient Global Optimization for Conceptual Rainfall-Runoff Models, Water Resour. Res., 28(4), 1015–1031, doi:10.1029/91wr02985, 1992.

---

## Author Comment (AC2) · 26 Jul 2016

Italicized text: Reviewer's comment

AR: Authors' response

**Comments to reviewer #2**

*The manuscript titled Combined assimilation of stream flow and snow water equivalent for mid-term ensemble stream flow forecasts in snow-dominated regions is an attempt to apply the Ensemble Kalman Filter to the application of stream flow prediction in regions where the majority of the water involved comes from snow.*

*The manuscript undertakes a set of studies, the first is to ascertain which of the 7 possible state variables they consider are sensitive to the assimilation of the 3 observation types they consider. They indicate that snow cover appears to not be an important factor in the forecasts that they seek.*

*The manuscript is well written and upon a second reading easy to follow. However, i do have a couple of points that need to be addressed before i can sign off on publication*

*Major Comments:*

*1) My first concern relates to the generation of the ensemble perturbations and the perturbations to the observations. You indicate that you use Gamma, lognormal of beta distributions yet the EnKF is highly reliant on these perturbations, and hence the errors being Gaussian distributed. My query, and i am requesting graphs of these, is to see the distributions plots for the distributions that you mentions with the parameters in the manuscript. My hunch is that these will look quite close to a Gaussian distribution of some form and as such is why you obtain the results, which are great results, but it could be misleading to have these distributions when really they are close to a Gaussian.*

AR: Many studies skip the controlled experiment and go straight to the assimilation of real data, but we feel this is not the ideal approach. As a first step to test DA potential on streamflow predictions, a near-ideal framework should be constructed in order to reduce the number "outside variables" that can influence results and mislead the analysis. Since the EnKF is used, all perturbations should have a normal distribution in order to obtain optimal results. However, observations like the ones used in our study have physical limits that cannot be breached (eg: SWE must be >= 0). Alternatives approaches must be used. In our study, we decided to use different distributions that 1) resemble normal ones when the mean is away from the limits, 2) prevent any violation of the physical limits of the variable and 3) are unbiased. The distributions introduced in the manuscript fit that description.

Examples of beta distributions obtained using the parameters presented in the manuscript, along with their analogous normal distributions obtained using the same mean and variance, are shown below (Fig. 1a of this response, has been added as Fig.4a in the revised manuscript). The variance is defined as a function of the mean in such a way that it is largest at 0.5 and smallest at the extremes in an attempt to reflect MODIS SCA retrieval's greater uncertainty during the transition periods when patchy snow is

prominent. The beta distribution also prevents perturbations from violating the physical limits of the variable, which the normal distribution cannot guarantee. A visual glance at the graphs shows that the beta distributions resemble normal distributions most near 0.5 and deviate more as the mean gets closer to the extremes.

In a similar fashion, examples of gamma and lognormal distributions using the parameters (variance values) described in the manuscript are show in Fig. 1b of this response (has been added as Fig.4b in the revised manuscript). A mean of 1 has been used for all examples for an easier comparison between examples, but the distributions themselves are visually independent of the mean since the variance is defined as directly proportional to the mean. The distributions resemble most a normal one for small variances relative to the mean, but important deviations can be noted as the relative variance increases due to the lower limit (zero) imposed on gamma and lognormal distributions. The variances shown are the ones used to perturb SWE and streamflow observations, as well as meteorological input.

While those distributions may or may not reflect entirely the real error of the observations and meteorological input, therefore yielding "optimistic" results, we feel they are a good compromise between the "normality" required by the EnKF and physical limits of the variables. As future works, one could investigate other distributions, introduce a bias, etc. and analyse their impact, but this falls outside the scope of the current manuscript.

The assumptions and limitations of the method are stated in the conclusion. In order to make it clearer that the experiment is near ideal, an additional sentences has been added in the revised manuscript.

[Figure]

Figure 1. Examples of a) beta and b) gamma and lognormal distribution compared with their analogous normal distributions using the same mean and variance.

*2) You need to provide a better justification to the use of these distributions on page 7.*

AR: A paragraph has been added in the revised manuscript to justify the use of non-normal distributions. Essentially, it is to satisfy hard boundaries on observations and meteorological input. Since all observations and meteorological input in the study have limits (ex: range between [0 infinity] or [0 1]), adding a normally distributed perturbation can mean those limits are sometimes exceeded. A simple way

to get around this problem might be to set all exceeding values at the boundary (ex : all negative values set to zero). However, this introduces a bias, which is another and likely bigger problem. Other distributions were therefore used to satisfy the physical limits, while keeping some visual similarities with a normal distribution if possible.

*3) You need to rewrite the paragraph starting on page 6 at line 14 as it is confusing as it would appear that it looks like you are referring to equations.*

AR: As suggested by another reviewer, the word "step" has been added before each number in parentheses to avoid confusion.

*4) The statement on page 17, line 30 does not make sense and is confusing about the need for linear relationships which you really should have with the EnKF.*

AR: The sentence is a relic of a previous formulation and has been removed.

*5) On page 9 you are finishing the details about the localization but i am concerned that because you achieve this wrt the true state that this may not be the case in the real data situation and you need some sort of disclaimer here as you are kind of using the true localization which would not be the case in reality.*

AR: This was meant to be discussed in the conclusion, in the paragraph explaining the assumptions and limitations of the results. This seems like a more appropriate section in the manuscript than the experimental design since 1) the results are not known yet in the experimental design section and 2) it allows us to generalize all results as valid within the synthetic limitations of the experiment. This avoids repetition since it applies to all steps, not only the covariance localization. An additional sentence has been added in the conclusion to explicitly state that dependency.

*Minor comments:*

*1) Page 3, line 27 remove to*

AR: This has been corrected in the revised manuscript.

*2) Page 5, line 17, you mention the gain yet you have not defined it.*

AR: The whole paragraph has been moved to the end of the section, as well as partly merged with the (previously) last paragraph and reworked to avoid repetition.

*3) Page 11, line 33, remove the first that*

AR: This has been corrected in the revised manuscript.

*4) Page 15, line 15, sits is not a very scientific way to describe where the site is.*

AR: The expression "sits at 682 mm" does not refer to the physical location of the site, but mean annual maximum SWE as described earlier in the sentence. Nonetheless, this has been rephrased to "contains a mean annual maximum SWE of 682 mm"

*5) General comment. you use both Gaussian and normal please be consistent and only use one of them*

AR: This has been corrected in the revised manuscript by using "normal" throughout the manuscript.

---

## Author Comment (AC3) · 26 Jul 2016

Italicized text: Reviewer's comment

AR: Authors' response

**Comments to reviewer #3 (Kevin He)**

*The authors present a set of synthetic experiments in assessing the potential added value of assimilating streamflow, SWE, SCA (via EnKF) into the CEQUEAU model in short- to medium-range streamflow forecasting at the Nechako watershed located in BC, Canada. Results indicate that streamflow DA and SWE DA lead to improvements in short-term forecast and medium-term forecast (during snow melt period), respectively. Assimilation of streamflow and SWE simultaneously yields even better results at both scales. However, assimilating SCA does not show any benefit. Overall, the methodology and results are sound and meaningful, yet not innovative. The paper is very well written and organized. I think it will be of interest to the readership of HESS.*

*My major comment is that, from the perspective of water resources management, the bias of the mean (or median) ESP forecast is typically an important factor considered in water-related decision making (e.g., water supply allocation, reservoir release/hydropower generation schedule, among others). In light of this, when assessing EPS forecast skill, the bias is normally analyzed.*

AR: We have been working closely with Rio Tinto and they, like many other water resources managers, also consider the bias to be an important metric (maybe even the most important), especially during the melt period. While this has been computed for all scenarios, as with many other metrics, we felt this actually added little to the discussion that did not justify the doubling of the number of figures for forecasts. This is because we have generated bias-free synthetic observations and meteorological input. This approach will likely always result in a nonzero bias due to the non-linearity of the hydrological model (e.g. two distributions of the same amount of rain can lead to different quantities of cumulated streamflow due to evapotranspiration, etc.) and the finite period used in the study, but it should ideally be very small.

In our case, average streamflow bias is less than 1 % for the open loop compared with the true state. Zooming on each year at a time, simulated and true cumulated streamflow difference oscillates around 5 % on average over the 10 years considered. This is mainly why the assimilation of SWE can lead to some improvement. If bias was always perfect, adding or removing snow would not lead to improvements. Although this bias value is improved in various ways with data assimilation, the window for improving bias remains small however. For the real world Nechako basin, average bias is estimated at around 20 %.

In current works that has not yet been published, biased precipitations are purposely added to test the robustness of the approach. Real data assimilation has also been evaluated. In both of those cases, bias becomes a central part of the results as it is significantly far from null to begin with (there is something to potentially improve upon). As it is currently however, we feel that adding figures of bias would contribute very little to the discussion due to the near-ideal framework used in the synthetic experiment.

*In the case of this study, the score MSSS is applied in the sensitivity analysis part (Figures 4, 6 and 7) but not the forecast part (Figures 8-12). The relevant results should be added (either in tabular or graphic form) and discussed.*

AR: The use of MSSS during the sensitivity analysis part and CRPSS during the forecast part was done to facilitate comparisons between other similar studies. The MSSS and CRPSS are relatively similar metrics used in different contexts. The mean square error (non-normalized version of the MSSS) is often used as a metric duration calibration or comparison between two curves, while the CRPS is mostly used to evaluate ESPs since it is adapted to ensembles. In the limit where an ensemble reaches 1 member, the CRPS simply becomes the mean absolute error. Although it is not exactly the same as the mean square error, the two carry much of the same information for non-ensemble curves. For ensembles, one should use the mean or median to compute the MSSS, which could yield very different results than the CRPSS since the latter is sensitive to the precision of the ensemble and not only its accuracy. However, in our case, the two respond in a very similar fashion as can been seen in fig. 1 (below) in this response. Similar graphs can be generated for all comparisons made using the CRPSS, but we believe this would add very little to the discussion.

[Figure]

**Figure 1. CRPSS and MSSS relative to the true state for a sample of forecasts.**

*My minor comments include 1) the authors need to be clear about how often the forecasts are issued (every day, once a week, or once per month in the study period from 8/15/1990 to 8/14/2000). If it is once a month, the authors need to discuss the sample size issue (10 years) when discussing the skill scores;*

AR: This has been clarified in a new section (3.1.3) of the revised manuscript dedicated to ensemble streamflow predictions.

*2) Line 7 of Page 2, "Franz" should be "Franz et al.";*

AR: This has been corrected in the revised manuscript.

*3) Lines 26-27 of Page 3, august should be August; delete "to";*

AR: This has been corrected in the revised manuscript.

*4) Line 8 of Page 4, (Fig. 2), Army Corps of Engineers;*

AR: This has been corrected in the revised manuscript.

*5) Lines 14-23 of Page 6, other than use (1), (2),. . ., it would less confusing when using (#1), (#2), . . ., or (Step 1), (Step 2), etc. ;*

AR: This has been modified in the revised manuscript by using the term "step" before each number.

*6) Line 5 of Page 7, delete "to";*

AR: This is a typo and has been replaced with "the".

*7) Line 6 of Page 8, modify "than";*

AR: This has been corrected in the revised manuscript by removing that part of the sentence. This was done in rearrangement of the sections to split the information relating to the generation of synthetic observations and observation ensembles.

*8) Lines 20-22 of Page 8, rework on the sentence;*

AR: The sentence has been reworked in the revised manuscript to:

"This approach avoids introducing a systematic bias when assimilating SCA values at 0 or 100 %. When values are at 0 % (or 100 %), perturbations can only introduce higher (or lower) values in order to remain within the physical limits of the observations. This approach also gives the observations a greater uncertainty during the transition periods when SCA, which loosely follows the greater uncertainty attributed to MODIS observations over the same period (Hall and Riggs, 2007)."

*9) Line 12 of Page 10, change MSS to MSSS;*

AR: This has been corrected in the revised manuscript.

*10) Line 20 of Page 10, in order to.*

AR: This has been corrected in the revised manuscript.

---

## Author Comment (AC4) · 6 Sep 2016

**Table 1: Overview of multivariate DA scenarios.**

| Method | Streamflow DA updates: | | SWE DA updates: | |
| --- | --- | --- | --- | --- |
| | VOL + GWL? | SWE? | VOL + GWL? | SWE? |
| VG-S | yes | no | no | yes |
| VG2-S | yes | no | yes | yes |
| VG-S2 | yes | yes | no | yes |
| VG2-S2 | yes | yes | yes | yes |

---

## Author Response (AR2)

**Authors' response for the manuscript "Combined assimilation of streamflow and snow water equivalent for mid-term ensemble streamflow forecasts in snow-dominated regions" by J. M. Bergeron, M. Trudel and R. Leconte**

We would like to thank the referee for providing constructive comments for the manuscript. The changes introduced since the last submission are the following:

1) A light rephrasing of three sentences for clarity purposes
2) The correction of a mix up in the legend of a table

These changes can be seen in the marked-up version of the manuscript with changes tracked below, following the point-by-point response to each of the referees.

Italicized text: Referee's comment

AR: Authors' response

**Comments to Referee #1**

*I was happy with the inclusion of the plots to highlight that the non-Gaussian distributions where close to a Gaussian such that the ENKF would at least be somewhat applicable to this problem. There are a couple of statements that are made that need to be addressed that are not quite true with respect to the ENKF and data assimilation in general but apart from that i am happy for the manuscript to be accepted subject to these minor changes.*

*1) In the abstract you say that overall the variables most closely linearly linked to the observations are the ones worth considering ..., this is only a restriction really for the ENKF, the hybrid and variational systems do not require this restriction, so this correction needs to be emphasized. It is also possible to use the extended version of the ENKF for nonlinear observation operators as well, so this restriction really only applies to the ENKF.*

*2) Page 6, line 19, you have known twice in this sentence and it does not make sense, re-write.*

*3) Page 7, line 28, the sentence appears not to be finished.*

AR: These sentences have been modified in the revised manuscript.

[revised manuscript text omitted]